# Expressive Power of Randomized Signature

**Christa Cuchiero**
University of Vienna, Department of Statistics and Operations Research
Kolingasse 14-16, 1090 Wien, Austria
`christa.cuchiero@univie.ac.at`

**Lukas Gonon**
Department of Mathematics, University of Munich
Theresienstraße 39, 80333 Munich, Germany
`gonon@math.lmu.de`

**Lyudmila Grigoryeva**
Department of Statistics, University of Warwick
Coventry, CV4 7AL, UK
`Lyudmila.Grigoryeva@warwick.ac.uk`

**Juan- Pablo Ortega**
Division of Mathematical Sciences, Nanyang Technological University
Singapore 637371
`Juan-Pablo.Ortega@ntu.edu.sg`

**Josef Teichmann**
Departement Mathematik, ETH Zürich
Rämistrasse 101, CH-8092 Zürich, Switzerland
`jteichma@math.ethz.ch`

## Abstract

We consider the question whether the time evolution of controlled differential equations on general state spaces can be arbitrarily well approximated by (regularized) regressions on features generated themselves through randomly chosen dynamical systems of moderately high dimension. On the one hand this is motivated by paradigms of reservoir computing, on the other hand by ideas from rough path theory and compressed sensing. Appropriately interpreted this yields provable approximation and generalization results for generic dynamical systems by regressions on states of random, otherwise untrained dynamical systems, which usually are approximated by recurrent or LSTM networks. The results have important implications for transfer learning and energy efficiency of training.

We apply methods from rough path theory, convenient analysis, non-commutative algebra and the Johnson-Lindenstrauss Lemma to prove the approximation results.

## 1   Introduction

In many areas of theoretical or practical interest, the quantitative understanding of dynamics in terms of initial values, local characteristics and noisy controls is pivotal. Classical approaches provide numerical algorithms to approximate dynamics given by such characteristics, where the quality of

35th Conference on Neural Information Processing Systems (NeurIPS 2021), Sydney, Australia.

approximation depends on the regularity of the local characteristics and on the nature of the noise. A bottleneck of this approach, which essentially goes back to Isaac Newton is the determination of characteristics from data. Machine-learning technologies based on universal approximation theorems follow a different approach to this problem: on a sufficiently rich training data set the map from initial values and control inputs to observed dynamics is learned directly without building an intermediate model upon its local characteristics. To learn such input-output maps one often applies recurrent neural networks, LSTMs or neural ODEs, which are flexible enough to approximate all sorts of dynamics but notoriously difficult to train.

Here the paradigm of reservoir computing, which – in different forms – appears in several areas of machine learning, enters the stage: split the input-output map into a generic part (the *reservoir*), which is *not or only in small parts* trained, and a readout part, which is accurately trained and often linear. In many cases, reservoirs can be realized physically, whence additionally ultrafast evaluations are possible, and learning the readout layer is often a simple (regularized) regression. Even though this approach seems surprising at first it has many well known instances, even in theoretical considerations, e.g. rough path theory, where a far reaching analysis of input-output systems is laid down.

Consider a controlled ordinary differential equation (CODE)

$$dY_t = \sum_{i=0}^{d} V_i(Y_t) du_t^i, \ Y_0 = y \in \mathbb{R}^N$$

for some smooth vector fields $V_i : \mathbb{R}^N \to \mathbb{R}^N$, $i = 0, \ldots, d$ and $d$ smooth control curves $u^i$. We place ourselves in the situation that we observe $u$ and $Y$, but do not have access to the vector fields $V^i$. The goal is to learn the dynamics and to simulate from it conditional on the controls $u$.

Whence we aim to describe the map

(input control $u$) $\mapsto$ (solution trajectory $Y$),

which is obviously a complicated, non-linear map of delicate regularity. A non-commutative version of Taylor's expansion provides a first step towards reservoirs here: indeed we can split (asymptotically in time) the map $(u_t)_{0 \leq t \leq T} \mapsto (Y_t)_{0 \leq t \leq T}$ into two parts:

- A linear system in the free algebra with $d$ generators

$$d\,\mathrm{Sig}_t = \sum_{i=0}^{d} \mathrm{Sig}_t \, e_i u_t^i, \ \mathrm{Sig}_0 = 1.$$

  This is an infinite dimensional system (called, e.g., the *signature process*) whose solution is just the collection of all iterated integrals of components of $u$, and which does *not* depend on the specific dynamics (hence this process is the *reservoir*).

- A linear map $\mathrm{Sig} \mapsto W \, \mathrm{Sig}$ which is actually trained (e.g. a linear regression!) and which is chosen to explain $(Y_t)_{0 \leq t \leq T}$ as good as possible (the *readout* map). In the present case we can, given the vector fields driving $Y$, calculate explicitly the coefficients of the readout map $W$ (possibly with regularization needed).

This well known split is not fully satisfying from the point of view of reservoir computing, since it is difficult, for a given control path $u$, to calculate the precise values of the reservoir $\mathrm{Sig}$. Indeed the reservoir system is an infinite dimensional dynamical system on a free algebra with all numerical complications arising in such situations. This is actually in sharp contrast to reservoir computing at work, where reservoirs are easy to evaluate. At this point we do *not* apply generic numerical methods to approximate the reservoir but rather compress the information of signature through a random projection

$$\pi : \mathbb{A}_d \to \mathbb{R}^k,$$

whose image can be surprisingly approximated by a dynamical system on $\mathbb{R}^k$, which has random characteristics. This is due to the fact that the characteristics of the signature process have a nilpotent structure, whence its random projections look by the central limit theorems almost like random matrices. The random projections are of course chosen according to the Johnson-Lindenstrauss lemma.

Therefore we can split (asymptotically in time), now fully in line with reservoir computing, the map $(u_t)_{0 \leq t \leq T} \mapsto (Y_t)_{0 \leq t \leq T}$ into two parts:

- A locally linear system on $\mathbb{R}^k$

$$dX_t = \sum_{i=0}^{d} \sigma(A_i X_t + b_i) \circ dB_t^i, \; Y_0 \in \mathbb{R}^k \, ,$$

where $\sigma$ is a componentwise applied activation function $\sigma : \mathbb{R} \to \mathbb{R}$ (notice that we explicitly allow the identity as a possible choice here) and $A_i$, $b_i$ are appropriately chosen random matrices or vectors (often just by independently sampling from $N(0,1)$). This is a finite dimensional system (the *reservoir*), which we call randomized signature process and which, again, does *not* depend on the specific dynamics.

- A linear map $X \mapsto WX$ which is actually trained (e.g. a linear regression) to explain $(Y_t)_{0 \le t \le T}$ as good as possible (the *readout* map).

Still we are able to prove generalization bounds for this approximation and we can observe them in an impressive way.

This also yields a fresh view on describing dynamics driven by, e.g., smooth controls $u$: one can either describe a dynamics by providing its local characteristics, i.e. the vector fields $V_1, \ldots, V_d$, or, in view of the above sketch, one can describe a dynamics by providing a reservoir, e.g. randomized signature, and the regression coefficients $W$. It is impressive that it is a linear task to learn $W$ from the (high-frequency) observation of one trajectory $Y$ together with the control $u$. This actually leads to a fascinating econometric approach which will be investigated in subsequent work.

## 2 Controlled ordinary differential equations on convenient vector spaces

Let $E$ be a convenient space, e.g. $E = \mathbb{R}^N$ (see Appendix A). Consider a controlled ordinary differential equation (CODE), i.e.

$$X_t = x + \sum_{i=1}^{d} \int_s^t V_i(X_r) du^i(r), \tag{1}$$

where $V_i : E \to E$ are some smooth vector fields on $E$. The *control* $u : \mathbb{R} \to \mathbb{R}^d$ is considered a smooth curve with values in $\mathbb{R}^d$. We refer to Section B for details on solutions to (1).

In order to understand algebra, analysis and geometry of iterated integrals, we have to consider the Hopf algebra $\mathbb{A}_d$ constructed by considering formal series generated by $d$ non-commutative indeterminates $e_1, \ldots, e_d$. A typical element $a \in \mathbb{A}_d$ is therefore written as

$$a = \sum_{k=0}^{\infty} \sum_{i_1, \ldots, i_k = 1}^{d} a_{i_1 \ldots i_k} e_{i_1} \cdots e_{i_k} \, .$$

We refer to Section B for details regarding the Hopf algebra structure of $\mathbb{A}_d$. For $i = 1, \ldots, d$ we define on $\mathbb{A}_d$ smooth vector fields

$$a \mapsto ae_i.$$

**Theorem 2.1.** *Let $u$ be a smooth control, then the controlled differential equation*

$$d\operatorname{Sig}_{s,t}(a) = \sum_{i=1}^{d} \operatorname{Sig}_{s,t}(a) e_i du^i(t) \, , \; \operatorname{Sig}_{s,s}(a) = a \tag{2}$$

*has a unique smooth evolution operator, called* signature *of $u$ and denoted by* Sig, *given by*

$$\operatorname{Sig}_{s,t}(a) = a \sum_{k=0}^{\infty} \sum_{i_1, \ldots, i_k = 1}^{d} \int_{s \le t_1 \le \cdots \le t_k \le t} du^{i_1}(t_1) \cdots du^{i_k}(t_k) \, e_{i_1} \cdots e_{i_k} \, . \tag{3}$$

*Remark* 2.2. Signature has many interesting properties which are encoded in the Hopf algebra $\mathbb{A}^d$. In particular signature is the solution of a time-homogenous dynamical system $\mathbb{A}^d$, which is crucial when it comes to applications in reservoir computing.

Denote by $\mathbb{A}_d^M$ the free nilpotent algebra with $d$ generators in which products of length more than $M$ vanish. The following splitting theorem is the precise link to reservoir computing and holds in this very general context of convenient vector spaces.

**Theorem 2.3.** *Let* Evol *be a smooth evolution operator on a convenient vector space $E$ which satisfies (where the time derivative is taken with respect to the forward variable $t$) a CODE* (1)

$$d\,\mathrm{Evol}_{s,t}(x) = \sum_{i=1}^d V_i(\mathrm{Evol}_{s,t}(x))du^i(t)\,.$$

*Then for any smooth (test) function $f : E \to \mathbb{R}$ and for every $M \geq 0$ there is a time-homogenous linear $W = W(V_1, \dots, V_d, f, M, x)$ from $\mathbb{A}_d^M$ to the real numbers $\mathbb{R}$ such that*

$$f\big(\mathrm{Evol}_{s,t}(x)\big) = W\big(\pi_M(\mathrm{Sig}_{s,t}(1))\big) + \mathcal{O}\big((t-s)^{M+1}\big)$$

*for $s \leq t$, where $\pi_M : \mathbb{A}_d \to \mathbb{A}_d^M$ is the canonical projection.*

*Remark* 2.4. In more literal terms: every controlled dynamics Evol can be split in a linear (readout) map $W$ and a (projection of) universal Sig dynamics.

# 3 The Johnson-Lindenstrauss lemma and randomly projected universal signature dynamics

It is the assertion of the Johnson-Lindenstrauss (JL) Lemma that for every $0 < \epsilon < 1$ an $N$ point set $Q$ in some arbitrary (scalar product) space $H$, can be embedded into a space $\mathbb{R}^k$, where $k = \frac{24 \log N}{3\epsilon^2 - 2\epsilon^3}$ in an almost isometric manner, i.e. there is a linear map $f : H \to \mathbb{R}^k$ such that

$$(1 - \epsilon)\|v_1 - v_2\|^2 \leq \|f(v_1) - f(v_2)\|^2 \leq (1 + \epsilon)\|v_1 - v_2\|^2$$

for all $v_1, v_2 \in Q$. It is remarkable that $f$ can be chosen randomly from a set of linear projection maps and the choice satisfies the desired requirements with high probability. The result is due to concentration of measure results in high dimensional spaces and has been discovered in the eighties.

We shall apply this remarkable result to obtain versions of signature in lower dimensional spaces which are possibly easier to evaluate. The geometry which we aim to preserve is the geometry of the defining controlled equation of signature: we therefore look for JL maps on $\mathbb{A}_d^M$ which preserve its geometry encoded in some set of directions $Q$.

In order to make this program work, we need a definition:

**Definition 3.1.** *Let $Q$ be any (finite or infinite) set of elements of norm one in $\mathbb{A}_d^M$ with $Q = -Q$. For $v \in \mathbb{A}_d^M$ we define the function*

$$\|v\|_Q := \inf \left\{ \sum_j |\lambda_j| \,\Big|\, \sum_j \lambda_j v_j = v \text{ and } v_j \in Q \right\}.$$

*We use the convention $\inf \emptyset = +\infty$ since the function is only finite on $\mathrm{span}(Q)$. Actually the function $\|.\|_Q$ behaves precisely like a norm on the span. Additionally $\|v\|_{Q_1} \geq \|v\|_{Q_2}$ for $Q_1 \subset Q_2$ and $\|v\|_Q \geq \|v\|$ for all sets $Q$ of elements of norm one.*

**Proposition 3.2.** *Fix $M \geq 1$, $\epsilon > 0$ and consider the free nilpotent algebra $\mathbb{A}_d^M$. Let $Q$ be any $N$ point set of vectors with norm one, then there is linear map $f : \mathbb{A}_d^M \to \mathbb{R}^k$ ($k$ being the above JL constant with $N$), such that*

$$\big|\langle v_1, v_2 - (f^* \circ f)(v_2)\rangle\big| \leq \epsilon,$$

*for all $v_1, v_2 \in Q$, where $f^* : \mathbb{R}^k \to \mathbb{A}_d^M$ denotes the adjoint map of $f$ with respect to the standard inner product on $\mathbb{R}^k$. In particular*

$$\big|\langle v_1, v_2 - (f^* \circ f)(v_2))\rangle\big| \leq \epsilon \|v_1\|_Q \|v_2\|_Q,$$

*for $v_1, v_2 \in \mathbb{A}_d^M$.*

By means of this special JL map associated to a point set $Q$, which we consider chosen relevant for the geometry, we can now project the vector fields in question without loosing too much information, we can solve the projected, reconstructed system and obtain – up to some time – a solution which is $\epsilon$-close to signature. By abuse of notation we shall write Sig in the sequel even though we mean the truncated version $\pi_M\big(\text{Sig}\,\big)(1)$ in $\mathbb{A}_d^M$.

The following theorem underlines the fact that randomized signature, as defined below, is as expressive as signature:

**Theorem 3.3.** *Let $u$ be a smooth control and $f$ the previously constructed JL map from $\mathbb{A}_d^M$ to $\mathbb{R}^k$. We denote by* r-Sig *the smooth evolution of*

$$dY_t = \sum_{i=1}^d \Big( \frac{1}{\sqrt{n}} f(f^*(Y_t)e_i) + (1 - \frac{1}{\sqrt{n}})f(\text{Sig}_{s,r}(1)e_i) \Big) du^i(t)\,,\ Y_0 \in \mathbb{R}^k\,,$$

*which is a controlled differential equation on $\mathbb{R}^k$. The natural number $n$ denotes the dimension of $\mathbb{A}_d^M$. Then*

$$\langle w, \text{Sig}_{s,t}(1) - f^*(\text{r-Sig}_{s,t}(Y_0))\rangle$$

$$\leq \big|\langle w, \text{Evol}_{s,t}(1 - f^*(Y_0))\rangle\big| + C\epsilon \sum_{i=1}^d \int_s^t \| \text{Evol}_{s,r}^* \, w\|_Q \| \text{Sig}_{s,r}\, e_i\|_Q \, dr\,,$$

*with constant $C = \sup_{s \leq r \leq t,\, i} \left| \frac{du^i(r)}{dr} \right|$, and for each $w \in \mathbb{A}_d^M$.*

It is fascinating that we can actually calculate approximately the vector fields which determine the dynamics of r-Sig by generic random elements.

**Theorem 3.4.** *For $M \to \infty$ the linear vector fields*

$$y \mapsto \frac{1}{\sqrt{n}} f(f^*(y)e_i)$$

*for $i = 1, \ldots, d$, are asymptotically normally distributed with independent entries. The time dependent bias terms*

$$(1 - \frac{1}{\sqrt{n}})f(\text{Sig}_{s,r}(1)e_i)\,.$$

*are as well asymptotically normally distributed with independent entries.*

## Acknowledgments and Disclosure of Funding

The authors are grateful for the support of the ETH Foundation.

## References

[1] C. Cuchiero, L. Gonon, L. Grigoryeva, J.-P. Ortega, and J. Teichmann. *Discrete-time signatures and randomness in reservoir computing*, IEEE Transactions on Neural Networks and Learning Systems (2021).

[2] Christa Cuchiero, Martin Larsson and Josef Teichmann, *Controlled ordinary differential equations on convenient spaces*, preprint, Arxiv (2019).

[3] Richard S. Hamilton, *The inverse function theorem of Nash and Moser*, Bull. Am. Math. Soc. **7** (1982), 65–222.

[4] Massimo Fornasier, Jan Haskovec, Jan Vybiral *Particle Systems and kinetic equations modeling interacting agents in high dimension*, SIAM J. Multiscale Modeling and Simulation, 2011

[5] Andreas Kriegl and Peter W. Michor, *The convenient setting for global analysis*, 'Surveys and Monographs 53', AMS, Providence, 1997.

[6] Serge Lang, *Fundamentals of differential geometry*, Graduate Texts in Mathematics 191, Springer, 1999.

# A Convenient Calculus

We resume the basic notions of convenient calculus (see [5] for all necessary details): a *convenient vector space* $E$ is a locally convex vector space such that all Mackey-Cauchy sequences converge. In particular all sequentially complete locally convex vector spaces are convenient. We denote by $E'$ the space of bounded linear functionals on $E$. On convenient vector spaces smooth curves, which are defined as usual, coincide with weakly smooth curves, i.e. $c : \mathbb{R} \to E$ is smooth if and only if $l \circ c \in C^\infty(\mathbb{R}, \mathbb{R})$ for all $l \in E'$. We define a new (in general finer) topology on $E$ to overcome the difficulty that there are obviously well behaved candidates for smooth mappings, which are not continuous: The $c^\infty$-topology is the final topology with respect to all smooth curves, so $U \subset E$ is open if the inverse image under any smooth curve to $E$ is open. A mapping $f : U \subset E \to F$ is called *smooth* if for all $c \in C^\infty(\mathbb{R}, E)$ the composition $f \circ c$ is a smooth curve to the convenient space $F$. This is the foundation of a consistent extension of classical analysis to the huge class of convenient vector spaces. Even on $\mathbb{R}^2$ it is not obvious that this definition of smoothness coincides with the classical one (see [5] for details on the history of convenient calculus). The main results are collected in the following theorem (for the proof see [5]).

**Theorem A.1.** *Let $E, F, G$ be convenient vector spaces, $U \subset E$, $V \subset F$ $c^\infty$-open, then we obtain:*

1. *Multilinear mappings are smooth if and only if they are bounded.*

2. *If $f : U \to F$ is smooth, then $\widehat{df} : U \times E \to F$ and $df : U \to L(E, F)$ are smooth, where*

$$df(x)(v) := \frac{d}{dt}|_{t=0} f(x + tv).$$

3. *The chain rule holds.*

4. *The vector space $C^\infty(U, F)$ of smooth mappings $f : U \to F$ is again a convenient vector space (inheritance property) with the following initial topology:*

$$C^\infty(U, F) \to \prod_{c \in C^\infty(\mathbb{R}, U)} C^\infty(\mathbb{R}, F) \to \prod_{c \in C^\infty(\mathbb{R}, U),\, \lambda \in F'} C^\infty(\mathbb{R}, \mathbb{R}).$$

5. *The exponential law holds, i.e.*

$$i : C^\infty(U, C^\infty(V, G)) \cong C^\infty(U \times V, G)$$

*is a linear diffeomorphism of convenient vector spaces. Usually we write $i(f) = \hat{f}$ and $i^{-1}(f) = \check{f}$.*

6. *The smooth uniform boundedness principle is valid: A linear mapping $f : E \to C^\infty(V, G)$ is smooth (bounded) if and only if $ev_v \circ f : E \to G$ is smooth for $v \in V$, where $ev_v : C^\infty(V, G) \to G$ denotes the evaluation at the point $v \in V$.*

7. *The smooth detection principle is valid: $f : U \to L(F, G)$ is smooth if and only if $ev_x \circ f : U \to G$ is smooth for $x \in F$ (This is a reformulation of the smooth uniform boundedness principle by cartesian closedness).*

8. *Taylor's formula is true, if one defines by applying cartesian closedness and obvious isomorphisms the multilinear-mapping-valued higher derivatives $d^n f : U \to L^n(E, F)$ of a smooth function $f \in C^\infty(U, F)$, more precisely for $x \in U$, $y \in E$ so that $[x, x + y] = \{x + sy | 0 \le s \le 1\} \subset U$ we have the formula*

$$f(y) = \sum_{i=0}^n \frac{1}{i!} d^i f(x) y^{(i)} + \int_0^1 \frac{(1-t)^n}{n!} d^{n+1} f(x + ty) (y^{(n+1)}) dt$$

*for all $n \in \mathbb{N}$.*

If not otherwise stated, $E$ and $F$ denote convenient spaces and $B$ a Banach space in what follows. We are looking for global solutions of ordinary differential equation with vector field $V : E \to E$, i.e. curves $x : \mathbb{R} \to E$ such that

$$dx(t) = V(x(t))dt\,, \ x(0) \in E\,,$$

where we pack – as usual – all sorts of time- or parameter dependence into $E$. Concerning differential equations, there are of course possible counterexamples on non-normable Fréchet spaces in all directions, which causes some problems in the foundations of differential geometry (see [5] for details). Nevertheless a useful generalization of the existence theorem for differential equations on Banach spaces is given by the following Banach map principle (see [3] for details, compare also [5], 32.14 for weaker results in a more general situation).

We shall, however, formulate one basic global existence and uniqueness result which is sufficient for the purposes of this article:

**Definition A.2.** *Given a convenient vector space E, a smooth map $V : E \to E$ is called a* tempered Banach map *if there are smooth (not necessarily linear) maps $R : E \to B$ and $Q : B \to E$ such that $V = Q \circ R$*

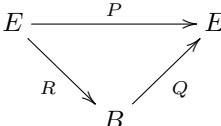

*where B is a Banach space.*

The space of Banach map vector fields $\mathcal{B}(E)$ is a $C^\infty(E, \mathbb{R})$-submodule of all vector fields $\mathfrak{X}(E)$. The following theorem is well known in the category of Fréchet spaces and due to Richard Hamilton.

**Theorem A.3** (Banach map principle)**.** *Let $V : E \to E$ be a Banach map, then $V$ admits a local flow on $E$.*

*Proof.* For the proof of the local version see [3], Theorem 5.6.3. The global statement follows from below. □

We are particularly interested in Banach maps perturbed by time-dependent and parameter-dependent linear generators of smooth semi-groups. Let $A(t, \theta) : E \to E$ be a bounded map on $E$ smoothly depending on 'time' $t$ and a one dimensional 'parameter' $\theta$. We assume that there is a smooth evolution (depending smoothly on all entries) of bounded linear operators $S_{s,t} = S_{s,t}^\theta \in L(E, E)$, for $s \leq t$, such that

$$\lim_{t \downarrow s} \frac{S_{s,t}^\theta - id}{t - s} = A(s, \theta)$$

which is a local solution operator for the time-dependent and parameter-dependent linear vector field $x \mapsto A(t, theta)x$.

Given a Banach map $V : E \times \mathbb{R} \times \mathbb{R} \to E$, we want to investigate the solutions of the perturbed initial value problem

$$\frac{d}{dt}x(t) = A^\theta(t)x(t) + V^\theta(t, x(t)), \quad x(0) = x_0.$$

**Theorem A.4.** *Let $E$ be a convenient vector space and let the smooth curve $(t, \theta) \mapsto A^\theta(t)$ be the time-dependent and parameter-dependent linear generator of a smooth evolution of linear maps $S_{s,t} \in L(E)$ of bounded linear operators on $E$ depending smoothly on all entries. Let $V : E \times \mathbb{R} \times \mathbb{R} \to E$ be a smoothly time-dependent and parameter-dependent Banach map. Then there is a unique smooth global evolution of solutions of the above initial value problem, in particular we have smooth dependence on the initial value and on the parameter $\theta$.*

*Proof.* See [2] and the references therein. □

# B   Controlled ordinary differential equations on convenient vector spaces

Let $E$ be a convenient space, see Section A for all details. Consider a controlled ordinary differential equation (CODE), i.e.

$$X_t = x + \sum_{i=1}^{d} \int_s^t V_i(X_r)du^i(r), \tag{4}$$

where $V_i : E \to E$ are some smooth vector fields on $E$ The *control* $u : \mathbb{R} \to \mathbb{R}^d$ is considered a smooth curve with values in $\mathbb{R}^d$.

For our purposes we shall only need local solvability in forward time direction, since then Theorem B.8 already holds. For classes of vector fields, where we have solutions for all times, see [2] and the references therein. We shall formulate a basic existence theorem:

**Theorem B.1.** *Let $E$ be a convenient vector space, $A$ a bounded linear generators of a smooth semi-group, $V_1, \ldots, V_d$ smooth Banach vector fields, and $u : \mathbb{R} \to \mathbb{R}^{d+1}$ a smooth control with $u^0$ increasing in time, then*

$$X_t = x + \int_s^t AX_r \, du^0(r) + \sum_{i=1}^d \int_s^t V_i(X_r) \, du^i(r) \tag{5}$$

*has a unique solution on $[0, \epsilon[$ for any initial value. Additionally the solution map*

$$(s, t, x, u) \mapsto \mathrm{Evol}^u_{s,t}(x) = \mathrm{Evol}_{s,t}(x)$$

*(for convenience we shall only denote the dependence on $u$ if necessary) satisfies the* evolution property

$$\mathrm{Evol}^u_{s,t} \circ \mathrm{Evol}^u_{r,s} = \mathrm{Evol}^u_{r,t} \tag{6}$$

*and $\mathrm{Evol}^u_{s,s}(x) = x$ for all $r, s, t$ and $x \in E$ where ever it is defined.*

*Proof.* Choose a curve of control $\theta \mapsto u_\theta \in C^\infty(\mathbb{R}, \mathbb{R}^d)$ and consider the differential equation

$$\frac{d}{dt}X_t = AX_t \frac{d}{dt}u^0_\theta(t) + \sum_{i=1}^d V_i(X_t) \frac{d}{dt}u^i_\theta(t), \quad X_0 = x.$$

We consider now $\theta$ as parameter and apply smooth dependence results of evolutions if vector fields are smooth. Since the operators $A$ generates a smooth semi-group,

$$t \mapsto A \frac{d}{dt}u^0_\theta(t)$$

generates a smooth evolution of bounded linear operators, see Section A, namely

$$S_{s,t}(x) = \mathrm{Fl}_{t-s}(x, s) = \exp\left(A_1(u^0_\theta(t) - u^0_\theta(s))\right)$$

for the respective initial value $x$ at time $s$. Then the perturbed equation, perturbed time-dependent and parameter-dependent vector fields,

$$(x, t, \theta) \mapsto \sum_{i=1}^d V_i(x) \frac{d}{dt}u^i_\theta(t)$$

has a unique solution smooth with respect to the initial values by Theorem A.4. By convenient analysis, Section A, it is clear that smoothness in $u$ is equivalent to showing smoothness in $\theta$ for all smooth curves of controls, whence the result follows. The evolution property follows from uniqueness and is part of the assertions of Theorem A.4. $\qquad\square$

**Definition B.2.** *Let $V : E \to E$ be a smooth vector field, and let $f : E \to \mathbb{R}$ be a smooth function, then we call*

$$Vf(x) = df(x) \bullet V(x)$$

*the transport operator associated to $V$, which maps smooth functions to smooth functions and determines $V$ uniquely.*

Let Evol be a smooth evolution operator on $c^\infty$-closed subset $A$ of a convenient vector space $E$ (or on a convenient manifold $M$ modeled on $E$), i.e. in particular the space $C^\infty(A; \mathbb{R})$ ($C^\infty(M; \mathbb{R})$ respectively) is well defined and determines the smooth curves into $A$ or $M$.

**Definition B.3.** *Let* Evol *be a smooth evolution on $A$ or $M$, then*

$$\text{T-Evol}_{s,t} : f \mapsto f \circ \mathrm{Evol}_{t,s}$$

*defines a linear evolution on the respective spaces of smooth functions, called the associated* transport evolution. *Indeed*

$$\text{T-Evol}_{s,t} \circ \text{T-Evol}_{r,s}(f) = f \circ \mathrm{Evol}_{s,r} \circ \mathrm{Evol}_{t,s} = f \circ \mathrm{Evol}_{t,r} = \text{T-Evol}_{r,t}(f).$$

**Theorem B.4.** *The smooth evolution operator* Evol *satisfies (the time derivative is taken with respect to the forward variable $t$) a controlled ordinary differential equation* (4)

$$d\,\mathrm{Evol}_{s,t}(x) = \sum_{i=1}^{d} V_i(\mathrm{Evol}_{s,t}(x))du^i(t)$$

*with initial value* $\mathrm{Evol}_{s,s}(x) = x \in E$. *Then the transport evolution satisfies*

$$d\,\mathrm{T\text{-}Evol}_{s,t}(f) = \sum_{i=1}^{d} V_i\,\mathrm{T\text{-}Evol}_{s,t}(f)du^i(t)\,.$$

*Proof.* By the chain rule and the evolution property we obtain

$$d\,\mathrm{Evol}_{s,t}(x) \bullet \Big( \sum_{i=1}^{d} V_i(x)\frac{d}{ds}u^i(s) \Big) = \sum_{i=1}^{d} V_i\big(\,\mathrm{Evol}_{s,t}(x)\big)\frac{d}{ds}u^i(s)$$

for all smooth evolution operators and for all real $s, t$. Indeed,

$$-\sum_{i=1}^{d} V_i(x)\frac{d}{ds}u^i(s) = \frac{d}{d\epsilon}|_{\epsilon=0}\big(\,\mathrm{Evol}_{s-\epsilon,s}(x)\big)$$

$$= \frac{d}{d\epsilon}|_{\epsilon=0}\big(\,\mathrm{Evol}_{t,s} \circ \mathrm{Evol}_{s-\epsilon,t}\,\big)(x)$$

$$= d\,\mathrm{Evol}_{t,s}(\mathrm{Evol}_{s,t}) \bullet \Big( -\sum_{i=1}^{d} V_i\big(\,\mathrm{Evol}_{s,t}(x)\big)\frac{d}{ds}u^i(s)\Big)$$

yields the result for $t \geq s$ where ever it is defined. $\qquad\square$

*Remark* B.5. For further details on derivatives of evolution operators see [2] and [5].

The following examples demonstrate how general the concept of a smooth evolution actually is:

*Example* B.6. The Laplace operator on an $n$ dimensional torus is a generator of a smooth semi-group on $E = C^\infty(\mathbb{T}^n)$. More generally, any strongly continuous semi-group on a Banach space $X$ defines a smooth semi-group on $\mathrm{dom}(A^\infty)$.

*Example* B.7. Vector fields of a shallow neural network type are Banach maps: let $l_1, \ldots, l_n : E \to \mathbb{R}$ be bounded linear maps, let $\phi : \mathbb{R} \to \mathbb{R}$ be a smooth map and $\alpha_1, \ldots, \alpha_n \in E$ be vectors, then

$$V(x) = \sum_{i=1}^{n} \alpha_i \phi\big(l_i(x) + \beta_i\big)$$

for some numbers $\beta_i \in \mathbb{R}$, $x \in E$, is a Banach map. Whence arbitrary neural network type perturbations of generators of strongly continuous semi-groups admit local solutions due to Theorem B.1.

Of utmost importance in our context is the following expansion theorem which allows in our smooth setting to expand solutions in terms of iterated integrals of the controls. The statements are well-known in control theory or rough path theory, still it is worth spelling it out in our convenient context.

**Theorem B.8.** *Let* Evol *be a smooth evolution operator on a convenient vector space $E$ which satisfies (again the time derivative is taken with respect to the forward variable $t$) a controlled ordinary differential equation* (4)

$$d\,\mathrm{Evol}_{s,t}(x) = \sum_{i=1}^{d} V_i(\mathrm{Evol}_{s,t}(x))du^i(t)$$

*then for any smooth function $f : E \to \mathbb{R}$*

$$f\big(\,\mathrm{Evol}_{s,t}(x)\big) = \sum_{k=0}^{M} \sum_{i_1,\ldots,u_k=1}^{d} V_{i_1}\cdots V_{i_k}f(x)\int_{s\leq t_1\leq\cdots\leq t_k\leq t} du^{i_1}(t_1)\cdots du^{i_k}(t_k)+ \quad (7)$$

$$+ R_M(s,t,f) \tag{8}$$

*with remainder term*

$$R_M(s,t,x,f) = \sum_{i_0,\ldots,u_M=1}^{d} \int_{s\leq t_1 \leq \cdots \leq t_{M+1}\leq t} V_{i_0}\cdots V_{i_k}f\big(\operatorname{Evol}_{s,t_0}(x)\big)du^{i_0}(t_0)\cdots du^{i_k}(t_M)$$

(9)

*holds true for all times $s \leq t$ and every natural number $M \geq 0$.*

*Remark* B.9. By convention the iterated integral over the empty set, which corresponds to the factor of $f$ is defined to be 1.

*Proof.* The proof is of an extraordinary simplicity and just consists iterating the defining integral equation along smooth functions: indeed take equation (4) and apply the chain rule for a function $f$, then

$$f\big(\operatorname{Evol}_{s,t}(x)\big) = f(x) + \int_s^t \sum_{i=1}^{d} V_i f\big(\operatorname{Evol}_{s,t_0}(x)\big)du^i(t_0),$$

which is just the asserted equation for $M = 0$. The equation describes what happens if Evol is inserted into a smooth function $f$, which is precisely what is needed inside the integral. Performing this step leads to the formula for $M = 1$. Notice that this is *not* what is done for Banach fixed point iteration.

The rest is done by induction. $\qquad\square$

*Remark* B.10. The formula holds verbatim (as long as integrals are defined) for continuous semi-martingales as well as for rough paths.

*Remark* B.11. In case $d = 1$, $V_1(x) = v$ for all $x \in E$, $u^1(t) = t$, the evolution is simply given by

$$\operatorname{Evol}_{s,t}(x) = x + v(t - s)$$

and the formula of Theorem B.8 is precisely Taylor's formula.

**Corollary B.12.** *In the assumptions of Theorem B.8 it holds that*

$$\big|R_M(s,t,x,f)\big| \leq \sup_{s\leq t_0\leq t}\big|V_{i_0}\cdots V_{i_k}f\big(\operatorname{Evol}_{s,t_0}(x)\big)\big| \sup_{s\leq r\leq t, i}\left|\frac{du^i(r)}{dr}\right|^{M+1}\frac{(t-s)^{M+1}}{(M+1)!},$$

*for all functions $f : E \to \mathbb{R}$ and for all $s \leq t$.*

In order to understand algebra, analysis and geometry of iterated integrals, we have to consider the following Hopf algebra, which can be easily topologized within the category of convenient vector spaces (proving again the practical aspect of this theory):

**Definition B.13.** *Consider the free algebra $\mathbb{A}_d$ of formal series generated by $d$ non-commutative indeterminates $e_1,\ldots,e_d$. A typical element $a \in \mathbb{A}_d$ is therefore written as*

$$a = \sum_{k=0}^{\infty} \sum_{i_1,\ldots,i_k=1}^{d} a_{i_1\ldots i_k}e_{i_1}\cdots e_{i_k},$$

*sums and products are defined in the natural way. We consider the complete locally convex topology making all projections $a \mapsto a_{i_1\ldots i_k}$ continuous on $\mathbb{A}_d$, hence a convenient vector space. We denote by $a_0$ the coefficient of $1$ and by $a_{\geq 1}$ the coefficients of products of length $k \geq 1$.*

*We can of course consider certain entire functions on $\mathbb{A}_d$:*

$$\exp(a_0 + a_{\geq 1}) = \exp(a_0)\sum_{i=0}^{\infty}\frac{1}{i!}a_{\geq 1}^i$$

*where the second series is understood as formal power series.*

*The co-product $\Delta$ is given by the unique algebra homomorphism with the property*

$$\Delta(e_i) = 1 \otimes e_i + e_i \otimes 1$$

*for $i = 1, \ldots, d$. The group like elements $G$ are precisely the exponential image of the Lie algebra $\mathfrak{g}$ generated by $e_1, \ldots, e_d$.*

*The antipodal map is given by extension of $e_i \mapsto -e_i$, which all together makes $\mathbb{A}_d$ Hopf algebra.*

*By abstract theory of Hopf algebras this means that the dual space, the free vector space on words in $d$ letters, is indeed again a Hopf algebra with the shuffle product and the concatenation co-product (and a respective antipodal map), in particular the space of linear maps restricted to the group $G$ is a point separating algebra. Whence any polynomial on $G$ can be written as a restriction to $G$ of a linear map. We shall see that elements from $G$ can actually be parametrized by iterated integrals, so polynomials of iterated integrals can be written as linear combinations of iterated integrals. This remarkable fact explains why the collection of iterated integrals actually universally linearizes dynamics driven by a control $u$.*

**Definition B.14.** *We define on $\mathbb{A}_d$ smooth vector fields*

$$a \mapsto a e_i$$

*for $i = 1, \ldots, d$.*

**Theorem B.15.** *Let $u$ be a smooth control, then the controlled differential equation*

$$d \operatorname{Sig}_{s,t}(a) = \sum_{i=1}^{d} \operatorname{Sig}_{s,t}(a) e_i du^i(t) , \ \operatorname{Sig}_{s,s}(a) = a \tag{10}$$

*has a unique smooth evolution operator, called* signature *of $u$ and denoted by* Sig, *given by*

$$\operatorname{Sig}_{s,t}(a) = a \sum_{k=0}^{\infty} \sum_{i_1, \ldots, u_k = 1}^{d} \int_{s \le t_1 \le \cdots \le t_k \le t} du^{i_1}(t_1) \cdots du^{i_k}(t_k) \, e_{i_1} \cdots e_{i_k} . \tag{11}$$

*Additionally $\operatorname{Sig}_{s,t}(1) \in G$ for all times $s, t$ and every $g \in G$ can be reached on any interval $[s, t]$ for $s < t$ by choosing an appropriate control $u$.*

*Proof.* A linear map is smooth if it maps smooth curves to smooth curves, hence the linear vector fields

$$a \mapsto a e_i$$

are smooth, since smoothness of a curve just means that the coordinate projections are smooth. Hence we have a well defined controlled differential equation on the convenient vector space $\mathbb{A}_d$.

We can identify the scale of ideals $\mathcal{I}_M$ in $\mathbb{A}_d$ of series whose expansion starts at $k = M + 1$. The factor algebra $\mathbb{A}_d^M$ is the free nilpotent algebra with $d$ generators, and of course finite dimensional since products of length more than $M$ vanish. We denote the canonical projection, which is an algebra homomorphism, by $\pi_M : \mathbb{A}_d \to \mathbb{A}_d^M$.

A curve $t \mapsto a(t) \in \mathbb{A}_d$ is smooth if and only if all its projections $t \mapsto \pi_M a(t) \in \mathbb{A}_d^M$ are smooth, for $M \ge 0$. We can now consider the projected differential equation, which is then a linear finite dimensional controlled differential equation, and solve it uniquely by applying $\pi_M$ to Formula (11). This in turn also proves that Sig is the unique smooth evolution operator solving Equation (10).

For the second statement we refer to [2] and the references therein: it is a consequence of the Chow-Rashevskii theorem of sub-riemannian geometry. $\qquad \square$

*Remark B.16.* Signature has many interesting properties which are encoded in the Hopf algebra $\mathbb{A}^d$. In particular signature is the solution of a time-homogenous dynamical system $\mathbb{A}^d$, which is crucial when it comes to applications in reservoir computing.

We conclude this section by the following splitting theorem, which is the precise link to reservoir computing and holds in this very general context of convenient vector spaces.

**Theorem B.17.** *Let* Evol *be a smooth evolution operator on a convenient vector space $E$ which satisfies (again the time derivative is taken with respect to the forward variable $t$) a controlled ordinary differential equation* (4)

$$d \operatorname{Evol}_{s,t}(x) = \sum_{i=1}^{d} V_i(\operatorname{Evol}_{s,t}(x)) du^i(t) .$$

*Then for any smooth (test) function $f : E \to \mathbb{R}$ and for every $M \geq 0$ there is a time-homogenous linear $W = W(V_1, \ldots, V_d, f, M, x)$ from $\mathbb{A}_d^M$ to the real numbers $\mathbb{R}$ such that*

$$f\big(\text{Evol}_{s,t}(x)\big) = W\big(\pi_M(\text{Sig}_{s,t}(1))\big) + \mathcal{O}\big((t-s)^{M+1}\big)$$

*for $s \leq t$.*

*Remark* B.18. In more literal terms: every controlled dynamics Evol can be split a linear (readout) map $W$ and a (projection of) a universal dynamics Sig.

*Proof.* For the proof combine the assertion of Theorem B.8, of Theorem B.15 and we define the linear map $W : \mathbb{A}_d^M \to \mathbb{R}$ via

$$W(e_{i_1} \cdots e_{i_k}) := V_{i_1} \cdots V_{i_k} f(x),$$

which is depending on the stated data and does not dependent on time. $\square$

**Corollary B.19.** *In case the remainder term in Theorem B.8 goes to zero as $M \to \infty$ the above expansion holds precisely, i.e. due to continuity*

$$f\big(\text{Evol}_{s,t}(x)\big) = W\big((\text{Sig}_{s,t}(1))\big).$$

*This is true in case of real analytic vector fields $V_1, \ldots, V_d$ and a real analytic function $f$ for times $s \leq t$ small enough.*

## C   Proofs

### C.1   Proof of Proposition 3.2

*Proof.* Consider $H = \mathbb{A}_d^M$ of dimension $m$. Then a JL map $f$ exists which almost preserves the distances of elements from $Q$ in $\mathbb{R}^k$. The JL lemma is due to the following estimate: take a matrix $k \times m$ matrix $A$ of independent standard normal random variables, then

$$\mathbb{P}\Big[(1-\epsilon)\|x\|^2 \leq \big\|\frac{1}{\sqrt{k}}Ax\big\|^2 \leq (1+\epsilon)\|x\|^2\Big] \leq 1 - 2\exp\big(-k\frac{\epsilon^2 - \epsilon^3}{4}\big)$$

for every $x \in \mathbb{R}^m$, $0 < \epsilon < 1$, and every $k, m \geq 1$. By taking unions for $x = v_1 - v_2$ over sets where the above bounds fail, for $v_1, v_2 \in Q$, one obtains a probability strictly less than one if $k$, $N$ and $\epsilon$ are properly chosen, e.g. with positive probability the bound holds. By polarization one obtains (notice that the length of elements in $Q$ is one)

$$\mathbb{P}\Big[\big|\langle v_1, v_2 \rangle - \langle \frac{1}{\sqrt{k}}Av_1, \frac{1}{\sqrt{k}}Av_2 \rangle\big| \geq \epsilon\Big] \leq 4\exp\big(-k\frac{\epsilon^2 - \epsilon^3}{4}\big).$$

Again by the same argument as above the desired assertion is obtained for $v_1, v_2 \in Q$. $\square$

### C.2   Proof of Theorem 3.3

*Proof.* Consider the controlled differential equation for signature, i.e.

$$d\,\text{Sig}_{s,t}(1) = \sum_{i=1}^{d} \text{Sig}_{s,t}\, e_i du^i(t), \ \ \text{Sig}_{s,s}(1) = 1$$

and look additionally at the controlled equation for randomized signature on $\mathbb{R}^k$

$$dY_t = (1 - \frac{1}{\sqrt{n}})\sum_{i=1}^{d} f(\text{Sig}_{s,r}(1)e_i)\, du^i(t) + \sum_{i=1}^{d} \frac{1}{\sqrt{n}}f(f^*(Y_t)e_i)du^i(t),$$

then we can consider the difference

$$\mathrm{Sig}_{s,t}(1) - f^*(Y_t)$$

$$= 1 - f^*(Y_0) + \sum_{i=1}^{d} \int_{s}^{t} \left( \mathrm{Sig}_{s,r}(1)e_i - \frac{1}{\sqrt{n}} f^*(f(f^*(Y_r)e_i))) du^i(r) - \right.$$

$$- \int_{s}^{t} (1 - \frac{1}{\sqrt{n}}) \sum_{i=1}^{d} f^*(f(\mathrm{Sig}_{s,r}(1)e_i)) \, du^i(r)$$

$$= 1 - f^*(Y_0) + \sum_{i=1}^{d} \int_{s}^{t} \frac{1}{\sqrt{n}} (f^* \circ f) \left( \mathrm{Sig}_{s,r}(1)e_i - f^*(Y_r)e_i \right) du^i(r) +$$

$$+ \sum_{i=1}^{d} \int_{s}^{t} \left( \mathrm{Sig}_{s,r}(1)e_i - (f^* \circ f) \left( \mathrm{Sig}_{s,r}(1)e_i \right) du^i(r) \right).$$

This can be solved by variation of constants, hence for every $w \in \mathbb{A}_d^M$ it holds by Proposition 3.2

$$\left| \langle w, \mathrm{Sig}_{s,t}(1) - f^* Y_t \rangle \right| \leq \left| \langle w, \mathrm{Evol}_{s,t}(1 - f^*(Y_0)) \rangle \right|$$

$$+ C \sum_{i=1}^{d} \int_{s}^{t} \left| \langle \mathrm{Evol}_{s,r}^* w, \mathrm{Sig}_{s,r}(1)e_i - (f^* \circ f) \left( \mathrm{Sig}_{s,r}(1)e_i \right) \rangle \right| dr$$

$$\leq \left| \langle w, \mathrm{Evol}_{s,t}(1 - f^*(Y_0)) \rangle \right| + C\epsilon \sum_{i=1}^{d} \int_{s}^{t} \left\| \mathrm{Evol}_{s,r}^* w \right\|_Q \left\| \mathrm{Sig}_{s,r} e_i \right\|_Q dr.$$

$\square$

## C.3  Proof of Theorem 3.4

*Proof.* For the proof see Theorem 3.7 in [1]. $\square$

