# OpenReview forum: "Expressive Power of Randomized Signature"
_NeurIPS.cc/2021/Workshop/DLDE — DLDE Workshop -- NeurIPS 2021 Poster_

### Official Review · Reviewer_jEiz · 2021-10-04

**Confidence:** 4

**Review:**

## Summary

A discussion on the expressivity of reservoir computing is provided. This is done by considering a particular class of reservoirs, namely randomised signatures, which are obtained by combining the signature of a process with the Johnson--Lindenstrauss Lemma.

## Discussion

The notation $f^*$ does not appear to be defined. (Have I missed it somewhere?) This makes it rather hard to understanding the precise nature of the theoretical statements being made.

The actual results obtained appear to be relatively unsurprising. The behaviour (and indeed approximation properties) of the signature, of CDEs, and of randomised projections are all relatively well-understood, so that these results may be combined makes sense.

It would be nice for the paper to have some more framing -- why is the problem being solved interesting? (Why should I, as a reader, care that this is a problem that has been solved?) I believe it is valuable to highlight "practical" connections (whether to numerical applications or to other theoretical applications). The text before Theorem 3.3 -- "randomized signature ... is as expressive as [sic] signature" -- appears to be the denouement, but this is scarcely highlighted at all.

All in all this seems to be essentially solid work. I think the only major weakness is that there is no discussion setting these results in context (to be discussed under the next heading of this review).

## Related work

There has been a substantial body of related work on this topic that goes entirely unaddressed. Even in a workshop paper it is important to set one's work in context. (Perhaps especially important, given the mix of backgrounds typical for the audience of an interdisciplinary workshop such as this one.) A few examples:

- Neural CDEs [1] [2] directly learn the vector fields of the equation of line 32. For example Appendix B of [1] demonstrates a very similar result to that shown here, namely a universality result for CDEs. (Broadly speaking signatures, reservoir computing, and RNNs/CDEs seem to be three different approaches to the same problem of learning dynamical systems: signatures have "universal" vector fields, reservoir computing has random vector fields, and RNNs/CDEs have learnt vector fields.)
- Neural RDEs [3] make use of more sophisticated tools from rough path theory -- the log-ODE method -- to derive an alternate numerical scheme by which numerical solutions may be obtained to a CDE.
- [4] offer a nice discussion, drawing on the same theory, to describe CDEs/RNNs as a kernel method. (e.g. to offer generalisation guarantees)

## Minor points

The equation of line 62 features what appears to be a Stratonovich integral against Brownian motion. This is contrast to the rest of the paper, which considers (Riemann--Stieltjes?) integrals against a control $u$. I assume this is just a copy-pasted typo.

What is the purpose of the "a" in equations (2) and (3)? In the sequel it appears to only ever take the value 1. (Which would be the conventional choice when defining the signature of a process.)

I find the notation of the paper to generally be a little clunky -- for example:

- Contrast Theorem 2.3 against Figure 3 of [3], which describes essentially the same thing.
- Line 127/128: why abuse notation? Why not just define Sig (or perhaps "Sig$^M$") as being the desired quantity?

## References

[1] Kidger et al. "Neural Controlled Differential Equations for Irregular Time Series" 2020 https://arxiv.org/abs/2005.08926

[2] Morrill et al. "Neural Controlled Differential Equations for Online Prediction Tasks" 2021 https://arxiv.org/abs/2106.11028

[3] Morrill et al. "Neural Rough Differential Equations for Long Time Series" 2021 https://arxiv.org/abs/2009.08295

[4] Fermanian et al. "Framing RNN as a kernel method: A neural ODE approach" 2021 https://arxiv.org/abs/2106.01202

**Score:**

3: Good paper

---

### Official Review · Reviewer_p8og · 2021-10-11

**Confidence:** 1

**Review:**

This paper seems to be theoretically sound, but is lacking the context and usage of this method.
I find the writing really hard to follow, and I think that the authors should try and find some practical examples where their approach is useful - this may increase the readability of the paper.  Also, although I'm not familiar with the particular field, it seems to me that the reference list is very short, containing only 6 references.

**Score:**

3: Good paper

---

### Official Review · Reviewer_H2vd · 2021-10-12

**Confidence:** 1

**Review:**

The authors suggest a method motivated by reservoir computing, rough path theory and non-commutative algebra. However, I find the formulation and the writing difficult to understand and follow, and suggest that the paper be revised to be more approachable for people with varied backgrounds. This could be the result of the extended abstract format, but nevertheless, I find it difficult to follow.

Otherwise the work seems to be sound and it is evident that substantial intellectual effort was put into the work.

**Score:**

3: Good paper

---

### Decision · Program_Chairs · 2021-10-15

**Decision:**

Accept (Poster)

**Comment:**

Reviews were positive and unanimously in favour of acceptance. Authors may wish to consider reviewers’ concerns in order to maximize the impact and accessibility of their poster at the workshop.